# Evolutionary Landscape of *SOX* Genes to Inform Genotype-to-Phenotype Relationships

**DOI:** 10.3390/genes14010222

**Published:** 2023-01-14

**Authors:** Adam Underwood, Daniel T Rasicci, David Hinds, Jackson T Mitchell, Jacob K Zieba, Joshua Mills, Nicholas E Arnold, Taylor W Cook, Mehdi Moustaqil, Yann Gambin, Emma Sierecki, Frank Fontaine, Sophie Vanderweele, Akansha S Das, William Cvammen, Olivia Sirpilla, Xavier Soehnlen, Kristen Bricker, Maram Alokaili, Morgan Green, Sadie Heeringa, Amy M Wilstermann, Thomas M. Freeland, Dinah Qutob, Amy Milsted, Ralf Jauch, Timothy J Triche, Connie M Krawczyk, Caleb P Bupp, Surender Rajasekaran, Mathias Francois, Jeremy W. Prokop

**Affiliations:** 1Division of Mathematics and Science, Walsh University, North Canton, OH 44720, USA; 2HudsonAlpha Institute for Biotechnology, Huntsville, AL 35806, USA; 3Department of Pediatrics and Human Development, College of Human Medicine, Michigan State University, Grand Rapids, MI 49503, USA; 4Single Molecule Science, Lowy Cancer Research Centre, The University of New South Wales, Sydney, NSW 2031, Australia; 5Institute for Molecular Bioscience, The University of Queensland, Brisbane, QLD 4072, Australia; 6Department of Chemistry, Grand Valley State University, Allendale, MI 49401, USA; 7Department of Biology, Calvin University, Grand Rapids, MI 49546, USA; 8School of Biomedical Sciences, Li Ka Shing Faculty of Medicine, The University of Hong Kong, Hong Kong SAR 518057, China; 9Center for Epigenetics, Van Andel Research Institute, Grand Rapids, MI 49503, USA; 10Department of Metabolism and Nutritional Programming, Van Andel Institute, Grand Rapids, MI 49503, USA; 11Division of Medical Genetics, Spectrum Health, Grand Rapids, MI 49503, USA; 12Office of Research, Spectrum Health, Grand Rapids, MI 49503, USA; 13The Centenary Institute, The University of Sydney, Royal Prince Alfred Hospital, Sydney, NSW 2006, Australia; 14Department of Pharmacology and Toxicology, Michigan State University, East Lansing, MI 48824, USA

**Keywords:** variant data integration, paralog mapping, SOX genes, developmental biology, transcription factor

## Abstract

The SOX transcription factor family is pivotal in controlling aspects of development. To identify genotype–phenotype relationships of SOX proteins, we performed a non-biased study of SOX using 1890 open-reading frame and 6667 amino acid sequences in combination with structural dynamics to interpret 3999 gnomAD, 485 ClinVar, 1174 Geno2MP, and 4313 COSMIC human variants. We identified, within the HMG (High Mobility Group)- box, twenty-seven amino acids with changes in multiple SOX proteins annotated to clinical pathologies. These sites were screened through Geno2MP medical phenotypes, revealing novel SOX15 R104G associated with musculature abnormality and SOX8 R159G with intellectual disability. Within gnomAD, SOX18 E137K (rs201931544), found within the HMG box of ~0.8% of Latinx individuals, is associated with seizures and neurological complications, potentially through blood–brain barrier alterations. A total of 56 highly conserved variants were found at sites outside the HMG-box, including several within the SOX2 HMG-box-flanking region with neurological associations, several in the SOX9 dimerization region associated with Campomelic Dysplasia, SOX14 K88R (rs199932938) flanking the HMG box associated with cardiovascular complications within European populations, and SOX7 A379V (rs143587868) within an SOXF conserved far C-terminal domain heterozygous in 0.716% of African individuals with associated eye phenotypes. This SOX data compilation builds a robust genotype-to-phenotype association for a gene family through more robust ortholog data integration.

## 1. Introduction

*SOX* (SRY-related HMG box) genes are involved in organ development and cell fate decisions [1]. These molecular switches can be improperly activated or dysregulated in numerous disease states. The first *SOX* gene discovered, the testes-determining factor *SRY*, was mapped by identifying sex-reversal variants for XY females [2,3]. Since that initial discovery, 19 additional *SOX* genes have been identified in humans, resulting in 20 genes with orthologs found in bilaterians [4,5]. All *SOX* genes encode a highly conserved high mobility group (HMG) box that binds and bends DNA, serving as an architectural and recruitment domain essential to gene regulation [6,7,8,9]. While the functional roles of some domains and regions outside the HMG box have been determined, the bulk of the work on SOX proteins has been carried out on the shared HMG box. This is primarily because these proteins are intrinsically disordered and difficult to validate functionally. With the rapid increase in sequenced genomes for rare, monogenic disorders, an informatics analysis of this gene family is needed now more than ever to prioritize investment in robust, comprehensive SOX gene knowledge relevant to human diseases. 

Pathogenic genomic variants have been identified in multiple *SOX* genes (Table 1 and Table 2). As indicated, *SRY* variants are associated with sex reversal [10]. *SOX9* variants also result in sex reversal, with a well-established role in Campomelic Dysplasia [11], a severe skeletal dysplasia. Genetic variants within *SOX2* are associated with anophthalmia and neural alterations [12,13], *SOX3* with altered pituitary development [14], *SOX10* with Waardenburg syndrome and Hirschsprung’s disease [15], *SOX11* with Coffin–Siris syndrome [16], and *SOX18* with Hypotrichosis–Lymphedema–Telangiectasia Syndrome (HLTRS) [17]. With the additive disease risks of variants in *SOX* genes, ongoing work has continued to define SOXopathies and summarize the phenotypic outcomes of modulating the gene family [18]. Recent work has suggested several cancer variants associated with gain-of-function variants in *SOX17* based on the screening of cancer genomic databases, furthering the reach of *SOX* variants into somatic variant risks [19]. 

In this study, we systematically assessed vertebrate SOX sequences to identify amino acid variations and functional regions. We assessed genomic variants in multiple public human-sequencing databases, utilizing deep conservation analysis to rank the potential impact of identified variants. Initial analysis compiling disease variants onto the sequences of HMG proteins showed that most rare disease variants occurred at sites conserved across the gene family, at structurally essential regions of the HMG box [20]. Not only have *SOX* genes been linked to numerous rare diseases, but mutations contributing to altered protein function and gene regulation have been associated with multiple forms of cancer [21]. Building on those studies, we provide an analysis of the genomic landscape for each of the 20 *SOX* genes across vertebrate evolution to assess human variants. 

## 2. Materials and Methods

### 2.1. Sequence and Structure Analysis

SOX gene/protein evolution was analyzed using our Sequence-to-Structure-to-Function analysis [22,23]. In short, open reading frame sequences (ORF) were isolated from NCBI so that only one sequence was used per species per gene. Sequences were aligned using the ClustalW [24] codon. Codon selection was called using a Muse–Gaut model of codon substitution [25] and dN/dS was called using HyPhy [26]. Any sequences with ambiguity or missing >9 nucleotides found in >90% of sequences for a gene were removed. All SOX sequences were aligned for the HMG box, and a phylogenetic tree was created using maximum likelihood with 1000 bootstrap analyses [27,28]. For each gene, the conservation score, a metric combining dN-dS and amino acid conservation [22], was placed on a 21-codon sliding window by adding the score of each position, with 10 upstream and 10 downstream. Domain annotations were identified using UniProt [29], and unknown domains were analyzed using ELM software [30]. Amino acid alignments were generated from NCBI gene orthologs [31] in November 2022, followed by alignment with COBALT [32]. The use of each amino acid across the 20 SOX proteins was called and colored onto the structure of SRY (PDB 1J46). Each protein was then modeled using 1J46 with YASARA homology modeling [33] and run for ten nanoseconds on molecular dynamic simulations (mds), as previously published in our extensive work on the HMG structures of SOX proteins [20,34]. The SOX18 E137K variant was assessed using PDB structure 4y60 [35] and processed for the wild type and variants over 20 nanoseconds of mds.

### 2.2. Genomic Variant Analysis

Genomic variants and phenotypic details were extracted from gnomAD [36], ClinVar [37], and Geno2MP [38] in November 2022. The Catalog of Somatic Mutations in Cancer (COSMIC) [39] was extracted in December 2022. Variants were called relative to the proximity to the HMG box, conservation in codon selection, conservation from amino acid sequence alignments, and frequency of the variant from each database. Further analysis of linkage disequilibrium using the phase 3 1000 genomes project was calculated using rAggr [40] with a specific focus on MXL and PEL populations. Additional disease-causing variants were curated from the literature (Table 1). 

### 2.3. SOX18 Culture Experiments

For luciferase assays, HeLa cells were seeded in triplicate at 6.6 × 10^3^ cells/cm^2^ in 24-well cassettes, cultured for 24 hr, and cotransfected using ViaFect. In each transfection reaction, we used 50 ng of effector plasmid (pHTC-Halo/SOX18, pHTC-Halo/SOX18-E137K, or empty pHTC-HaloTag vector), 500 ng of SOX-inducible luciferase reporter (as described previously [41]), and 1 ng of control vector (phRL-null Renilla). The cells were incubated for 30 hr and assessed using the Dual-Luciferase^®®^ Reporter Assay System measured on a GloMax^®®^ Luminometer. Each effector’s relative transcriptional activity was measured after normalizing Firefly luciferase to Renilla luciferase and then calculating the fold change relative to the empty effector plasmid. 

The Amplified Luminescent Proximity Homogeneous Assay (Alpha) screening for SOX18 was performed as previously described [34,42,43]. A STRING [44] network map of SOX18, RBPJ, and MEF2C was created using no more than ten partners in both shells one and two of the network. GO enrichment was then assessed on the STRING tool. 

**Table 1 genes-14-00222-t001:** SOX gene variants with potential functional outcomes extracted from the literature. Notes: ^a^ HMG box AA numbers are listed first based on UniProt and then listed based on Bowles/Koopman, where there is an offset of 2 bases in the annotations. References can be found in [10,17,45,46,47,48,49,50,51,52,53,54,55,56,57,58,59,60,61,62,63,64,65,66,67,68,69,70,71,72,73,74,75,76,77,78].

HMG Box AA ^a^	Sox AA	AA	NV	Gene	Disease	Reference	Data Source (accessed on 18 December 2022)
-	3	S	L	*SRY*	SRXY1	Gimelli et al., 2007	http://www.ncbi.nlm.nih.gov/pubmed/17063144
-	18	S	N	*SRY*	Partial SRXY1	Domenice et al., 1998	http://www.ncbi.nlm.nih.gov/pubmed/9521592
-	18	S	N	*SRY*	Tuner Syndrome	Canto et al., 2000	http://www.ncbi.nlm.nih.gov/pubmed/10843173
-	59	R	G	*SRY*	45,X/46,X psu dic (Y)	Fernandez et al., 2001	http://www.ncbi.nlm.nih.gov/pubmed/12215836
1/3	60	V	L	*SRY*	SRXY1	Vilain et al., 1992	http://www.ncbi.nlm.nih.gov/pubmed/1570829
1/3	60	V	L	*SRY*	SRXY1	Berta et al., 1990	http://www.ncbi.nlm.nih.gov/pubmed/2247149
1/3	60	V	A	*SRY*	SRXY1	Hiort et al., 1995	http://www.ncbi.nlm.nih.gov/pubmed/7776083
3/5	106	R	W	*SOX10*	WS4C	Chaoui et al., 2011	http://www.ncbi.nlm.nih.gov/pubmed/21898658
3/5	62	R	G	*SRY*	SRXY1	Affara et al., 1993	http://www.ncbi.nlm.nih.gov/pubmed/8353496
4/6	108	P	L	*SOX9*	CMD1	Meyer et al., 1997	http://www.ncbi.nlm.nih.gov/pubmed/9002675
5/7	64	M	I	*SRY*	SRXY1	Berta et al., 1990	http://www.ncbi.nlm.nih.gov/pubmed/2247149
5/7	64	M	R	*SRY*	SRXY1	Scherer et al., 1998	http://www.ncbi.nlm.nih.gov/pubmed/9678356
8/10	67	F	V	*SRY*	SRXY1	Scherer et al., 1998	http://www.ncbi.nlm.nih.gov/pubmed/9678356
8/10	112	F	L	*SOX9*	CMD1	Kwok et al., 1995	http://www.ncbi.nlm.nih.gov/pubmed/7485151
8/10	112	F	S	*SOX9*	CMD1	Goji et al., 1998	http://www.ncbi.nlm.nih.gov/pubmed/9452059
9/11	68	I	T	*SRY*	SRXY1	Haqq et al., 1994	http://www.ncbi.nlm.nih.gov/pubmed/7985018
9/11	113	M	T	*SOX9*	CMD1	Wada et al., 2009	http://www.ncbi.nlm.nih.gov/pubmed/19921652
9/11	113	M	V	*SOX9*	CMD1	Staffler et al., 2010	http://www.ncbi.nlm.nih.gov/pubmed/20513132
9/11	112	M	I	*SOX10*	WS2E/PCWH	Chaoui et al., 2011	http://www.ncbi.nlm.nih.gov/pubmed/21898658
11/13	95	W	R	*Sox18*	HLTS	Irrthum et al., 2003	http://www.ncbi.nlm.nih.gov/pubmed/12740761
15/17	119	A	V	*SOX9*	CMD1	Kwok et al., 1995	http://www.ncbi.nlm.nih.gov/pubmed/7485151
17/19	76	R	S	*SRY*	SRXY1	Imai et al., 1999	http://www.ncbi.nlm.nih.gov/pubmed/10670762
19/21	78	M	T	*SRY*	SRXY1	Affara et al., 1993	http://www.ncbi.nlm.nih.gov/pubmed/8353496
20/22	104	A	P	*Sox18*	HLTS	Irrthum et al., 2003	http://www.ncbi.nlm.nih.gov/pubmed/12740761
28/30	87	N	Y	*SRY*	SRXY1	Okuhara et al., 2000	http://www.ncbi.nlm.nih.gov/pubmed/10721678
28/30	131	N	H	*SOX10*	PCWH	Chaoui et al., 2011	http://www.ncbi.nlm.nih.gov/pubmed/21898658
30/32	89	E	K	*SRY*	SRXY1	Cunha et al., 2011	http://www.ncbi.nlm.nih.gov/pubmed/21344134
31/33	90	I	M	*SRY*	SRXY1	Hawkins et al., 1992a	http://www.ncbi.nlm.nih.gov/pubmed/1415266
31/33	90	I	M	*SRY*	SRXY1	Dork et al., 1998	http://www.ncbi.nlm.nih.gov/pubmed/9450909
31/33	90	I	M	*SRY*	SRXY1	Maier et al., 2003	http://www.ncbi.nlm.nih.gov/pubmed/12793612
32/34	91	S	G	*SRY*	SRXY1	Schmitt-Ney et al., 1995	http://www.ncbi.nlm.nih.gov/pubmed/7717397
33/35	92	K	M	*SRY*	SRXY1	Shahid et al, 2009	http://www.uniprot.org/uniprot/D0VTX3
35/37	94	L	W	*SRY*	Tuner Syndrome	Shahid et al., 2009	http://www.uniprot.org/uniprot/D0VTX0
36/38	95	G	E	*SRY*	SRXY1	Schaeffler et al., 2000	http://www.ncbi.nlm.nih.gov/pubmed/10852465
36/38	95	G	R	*SRY*	SRXY1	Hawkins et al., 1992b	http://www.ncbi.nlm.nih.gov/pubmed/1339396
39/41	143	W	R	*SOX9*	CMD1	Meyer et al., 1997	http://www.ncbi.nlm.nih.gov/pubmed/9002675
42/44	101	L	H	*SRY*	SRXY1	Braun et al., 1993	http://www.ncbi.nlm.nih.gov/pubmed/8447323
42/44	145	L	P	*SOX10*	WS4C	Chaoui et al., 2011	http://www.ncbi.nlm.nih.gov/pubmed/21898658
47/49	106	K	I	*SRY*	SRXY1	Hawkins et al., 1992a	http://www.ncbi.nlm.nih.gov/pubmed/1415266
47/49	150	K	N	*SOX10*	PCWH	Chaoui et al., 2011	http://www.ncbi.nlm.nih.gov/pubmed/21898658
48/50	152	R	P	*SOX9*	CMD1	Meyer et al., 1997	http://www.ncbi.nlm.nih.gov/pubmed/9002675
49/51	108	P	R	*SRY*	SRXY1	Jakubiczka et al., 1999	http://onlinelibrary.wiley.com/doi/10.1002/(SICI)1098-1004(1999)13:1%3C85::AID-HUMU16%3E3.0.CO;2-O/abstract
50/52	109	F	S	*SRY*	SRXY1	Jaeger et al., 1992	http://www.ncbi.nlm.nih.gov/pubmed/1483689
50/52	154	F	L	*SOX9*	CMD1	Preiss et al., 2000	http://www.ncbi.nlm.nih.gov/pubmed/11323423
54/56	113	A	T	*SRY*	SRXY1	Zeng et al., 1993	http://www.ncbi.nlm.nih.gov/pubmed/8105086
54/56	158	A	T	*SOX9*	CMD1	Preiss et al., 2000	http://www.ncbi.nlm.nih.gov/pubmed/11323423
54/56	157	A	V	*SOX10*	WS4C	Chaoui et al., 2011	http://www.ncbi.nlm.nih.gov/pubmed/21898658
58/60	161	R	H	*SOX10*	WS2E	Chaoui et al., 2011	http://www.ncbi.nlm.nih.gov/pubmed/21898658
59/61	118	A	P	*SRY*	SRXY1	Shahid et al, 2009	http://www.uniprot.org/uniprot/D0VTX2
61/63	165	H	Y	*SOX9*	CMD1	McDowall et al., 1999	http://www.ncbi.nlm.nih.gov/pubmed/10446171
61/63	165	H	Q	*SOX9*	CMD1	Staffler et al., 2010	http://www.ncbi.nlm.nih.gov/pubmed/20513132
66/68	170	P	R	*SOX9*	CMD1	Meyer et al., 1997	http://www.ncbi.nlm.nih.gov/pubmed/9002675
66/68	170	P	L	*SOX9*	CMD1	Wada et al., 2009	http://www.ncbi.nlm.nih.gov/pubmed/19921652
66/68	125	P	L	*SRY*	SRXY1	Schmitt-Ney et al., 1995	http://www.ncbi.nlm.nih.gov/pubmed/7717397
68/70	127	Y	C	*SRY*	SRXY1	Poulat et al., 1994	http://www.ncbi.nlm.nih.gov/pubmed/8019555
68/70	127	Y	F	*SRY*	SRXY1	Jordan et al., 2002	http://www.ncbi.nlm.nih.gov/pubmed/12107262
68/70	127	Y	I	*SRY*	SRXY1	Shahid et al., 2009	http://www.uniprot.org/uniprot/D0VTX2
69/71	173	K	E	*SOX9*	CMD1	Thong et al., 2000	http://www.ncbi.nlm.nih.gov/pubmed/10951468
71/73	174	Q	P	*SOX10*	PCWH	Chaoui et al., 2011	http://www.ncbi.nlm.nih.gov/pubmed/21898658
72/74	131	P	R	*SRY*	SRXY1	Lundberg et al., 1998	http://onlinelibrary.wiley.com/doi/10.1002/humu.13801101108/abstract
72/74	175	P	A	*SOX10*	PCWH	Chaoui et al., 2011	http://www.ncbi.nlm.nih.gov/pubmed/21898658
72/74	175	P	L	*SOX10*	PCWH	Chaoui et al., 2011	http://www.ncbi.nlm.nih.gov/pubmed/21898658
72/74	175	P	R	*SOX10*	PCWH	Chaoui et al., 2011	http://www.ncbi.nlm.nih.gov/pubmed/21898658
74/76	133	R	W	*SRY*	SRXY1	Affara et al., 1993	http://www.ncbi.nlm.nih.gov/pubmed/8353496

## 3. Results

### 3.1. Sequence Evolution of SOX Genes/Proteins

A total of 1890 open reading frame (ORF) sequences of vertebrates were curated for 20 SOX genes, averaging just under 100 species per gene analyzed. In addition, we utilized more up-to-date (as of November 2022) amino acid multiple-species comparisons with 6667 sequences averaging 333 species for each SOX protein. The ORF alignments of the SOX genes allowed for the analysis of selection throughout each of the genes. Following codon and amino acid analysis for each of the 20 genes, a sliding window metric [22] was applied to identify conserved domains and linear motifs within each gene normalized such that the highest motif had a value of 1 (Figure 1). Only the HMG domain was conserved for SOX1, SOX2, SOX3, SOX15, SOX21, SOX30, and SRY (Figure 1, red boxes). The HMG domain numbers were based on UniProt, where additional SOX-centric numbering used by others [5] was shifted by two amino acids. In general, most of the 485 ClinVar variants were within HMG domains, while the 3999 gnomAD and 1174 Geno2MP variants were mostly found outside of the HMG domain (Figure 1, Table 2). Several SOX genes had conserved domains in addition to the HMG box annotated in databases such as UniProt (Figure 1, magenta boxes), and several regions were conserved without previous annotation (Figure 1, gray boxes). There were also ClinVar annotated variants within these regions that required further analyses.

**Table 2 genes-14-00222-t002:** SOX genes with known disease associations and mapping data for genomic variants. In red at the bottom are averages or total numbers for various columns. AD—autosomal dominant; XL—X-linked; XLD—X-linked dominant; AR—autosomal recessive; NA—nucleic acid sequences; AA—amino acid sequences.

Gene	Group	Ensembl	NCBI	Missense Constraint	OMIM/ClinVar	Inheritance	AA	Species NA	Species AA	gnomAD	ClinVar	Geno2MP	COSMIC
*SOX1*	B	ENST00000330949	NP 005977	0.58	-	-	391	73	348	120	0	115	134
*SOX2*	B	ENST00000325404	NP 003097	2.12	Microphthalmia	AD	317	88	452	119	70	66	203
*SOX3*	B	ENST00000370536	NP 005625	2.21	Panhypopituitarism	XL	446	74	194	104	18	39	226
*SRY*	A	ENST00000383070	NP 003131	−0.14	sex reversal	XLD	204	30	57	25	26	5	25
*SOX14*	B	ENST00000306087	NP 004180	1.14	-	-	240	100	465	118	0	20	112
*SOX21*	B	ENST00000376945	NP 009015	1.74	-	-	276	104	388	81	0	60	79
*SOX4*	C	ENST00000244745	NP 003098	1.17	Coffin-Siris syndrome	AD	474	103	281	210	27	64	125
*SOX11*	C	ENST00000322002	NP 003099	2.26	Coffin-Siris syndrome	AD	441	122	420	166	47	38	282
*SOX12*	C	ENST00000342665	NP 008874	1.79	-	-	315	71	261	106	1	27	63
*SOX5*	D	ENST00000451604	NP 008871	3.15	Lamb-Shaffer syndrome	AD	763	89	445	264	46	51	480
*SOX6*	D	ENST00000528429	NP 001354802	1.94	Tolchin-Le Caignec syndrome	AD	828	101	438	365	17	86	380
*SOX13*	D	ENST00000367204	NP 005677	1.74	-	-	622	94	321	296	0	91	199
*SOX8*	E	ENST00000293894	NP 055402	0.89	-	-	446	143	301	254	3	98	183
*SOX9*	E	ENST00000245479	NP 000337	1.63	Campomelic dysplasia	AD	509	112	375	259	97	55	524
*SOX10*	E	ENST00000396884	NP 008872	2.77	PCWH syndrome	AD	466	100	434	176	100	44	198
*SOX7*	F	ENST0000030450	NP 113627	−0.74	-	-	388	149	444	420	2	81	204
*SOX17*	F	ENST00000297316	NP 071899	0.77	Vesicoureteral reflux	AD	414	100	227	225	18	63	424
*SOX18*	F	ENST00000340356	NP 060889	0.86	HLTS	AD/AR	384	102	287	151	9	75	93
*SOX15*	G	ENST00000250055	NP 008873	0.62	-	-	233	97	210	118	0	29	83
*SOX30*	H	ENST00000265007	NP 848511	0.78	Male infertility	AD	753	111	319	422	4	67	296
			* **Average** *	* **1** *		* **Total** *	* **8910** *	* **1963** *	* **6667** *	* **3999** *	* **485** *	* **1174** *	* **4313** *

### 3.2. Mapping SOX Variants 

Deep evolutionary analysis of the SOX genes has the potential to identify functional regions and amino acids, prioritizing variants of interest for future analyses. Advanced metrics to look at the general intolerance of mutations in genes across the entire genome, such as RVIS scores [79], suggest that SOX genes are generally tolerant to human variants, with a value around the 37th percentile of all genes (SOX2—45.64%; SOX4—31.22%; SOX5—5.30%; SOX6—4.93%; SOX7—22.47%; SOX8—76.58%; SOX9—9.40%; SOX10—27.70%; SOX11—33.83%; SOX13—61.79%; SOX14—40.18%; SOX15—48.78%; SOX17—34.82%; SOX30—73.37%). The RVIS (residual variation intolerance score) represents a value for how much a gene can tolerate variants, whereas most transcription factor scores suggest that they can tolerate changes. However, these metrics are biased based on the degree of domain and motif conservation relative to the size of the protein. The relative Z-scores of gnomAD variants (missense constraint, Table 2) computed based on the predicted number of variants based on all human genes also suggested a neutral level of SOX variants (average of 1). Therefore, a more systematic metric of SOX-gene-to-variant impact is needed. 

We identified 3623 unique amino acids with germline variant sites within SOX proteins in population genomics (gnomAD), clinical genomes with matched phenotype descriptions (Geno2MP), and clinical genomes annotated within ClinVar (Figure 2A). The largest number of these amino acids had only gnomAD annotations, while Geno2MP and the overlap of gnomAD and Geno2MP were identified as the next-largest groups. As the conservation analysis highlighted, the HMG box had a critical function for all SOX members. Only within the ClinVar database were variants found more often within the HMG-box (red, Figure 2A), with all other databases having variants rarely falling within the domain. 

Our two metrics of conservation, the codon selection score generated by open reading frame alignments and amino acid conservation using a higher number of species protein sequences, showed that the codon selection scores provided a higher stratification of values. In contrast, amino acid conservation had more variants annotated at conserved sites (Figure 2B). Therefore, we focused on the codon selection scores to further annotate variants.

Within ClinVar, the HMG box variants had a higher codon selection score with high enrichment for the amino acids with evidence of selective pressure (ranking from 1 to 2, Figure 2C). Within Geno2MP and gnomAD, while HMG box variants were rare, when a variant fell within the HMG domain, they did fall at sites of higher conservation than those outside the HMG domain. Further, the amino acids with a high codon selection score within gnomAD had very low allele frequencies relative to sites with low scores (Figure 2D), suggesting that common variants of SOX proteins were at sites with limited selection. Additionally, many of the HMG box variants of gnomAD had low allele frequency, except for a SOX18 E137K variant. Similarly, Geno2MP variants had a lower number of HPO profiles when the codon selection scores were higher, while those rare variants often fell within the HMG domain with higher predicted CADD scores (Figure 2E). This suggests that a more detailed analysis of the HMG domain across the SOX members would benefit our ability to interpret SOX variants.

### 3.3. Interpreting HMG Box Variants

We began with a detailed analysis of amino acid alignments and structural insights for the HMG box between different SOX members, scaling insights into the orthologous domain. All 1890 ORF SOX HMG box sequences were aligned, and the evolutionary relationships were compared. Most sequences clustered well into the 20 genes, further clustering into the previously identified subgroups of SOX (https://doi.org/10.6084/m9.figshare.14544063, accessed on 18 December 2022, and https://doi.org/10.6084/m9.figshare.14544219, accessed on 18 December 2022). Using the alignment data, we mapped the number of amino acids used throughout the 20 human SOX proteins onto the solved HMG-box protein structure (PDB file 1J46, Figure 3). There was a high clustering of singly used amino acids (conserved throughout) in the core three-helix bundle of the HMG box, where conserved amino acids can be observed (red) in a dense core of the three-helix packing. Amino acid usage away from the HMG packing and DNA contacts could also be found (Figure 3A). The most amino acids used at any site of the HMG box was seven, occurring at five different sites (Figure 3B).

The positions with only a single amino acid had high selective pressure (dN-dS metric, Figure 3C), except for M at position 5, W at 11, and W at 39. M/W used only a single codon and thus did not possess dN-dS selection metrics. In contrast, all other sites with a single amino acid were suggested based on dN-dS metrics to have high selective pressure throughout all 20 SOX genes. All the DNA contact positions had only a single amino acid used in all sequences (Figure 3D). They were significantly stabilized when DNA was added to all 20 proteins as determined by molecular dynamic simulations (Figure 3E). This suggests that all SOX HMG boxes recognized similar DNA sequences (such as AACAA) through the DNA binding domain. Any deviation in binding or target recognition was likely driven by protein-specific non-HMG flanking DNA interactions, protein–protein recruitment from regions outside DNA binding, dimerization, chromatin penetrance, or competition from other transcription factors within cellular environments, such as other HMG box or fork-head transcription factors.

Using the UniProt annotation of the HMG box, an analysis of relative HMG box positioning for all variants of ClinVar, Geno2MP, and gnomAD was conducted. This approach provides ortholog mapping insights where a clinical variant at an HMG box number could assist in further validating variants of uncertain biological impact. Flanking the HMG box (−2 to −10 and 76–100, where 1 is the first UniProt annotated amino acid of the HMG box), the amino acids had higher variability between SOX members (Figure 4A), a lower codon selection score (Figure 4B), and a lower amino acid conservation score (Figure 4C). Integrating ClinVar annotated variants and those within the literature (Table 1) identified many HMG box sites where multiple SOX genes had annotated pathologies connected to their variation (Figure 4D). Geno2MP (Figure 4E) and gnomAD variants (Figure 4F) were lower within the HMG box positions relative to those flanking the domain.

Various resources provide different numbering of HMG amino acids. Using the UniProt numbering position −1 (2 within the Bowles/Koopman numbering) was the first to have an average of over 90% conserved (93 ± 9%) in the 6667 amino acid sequences analyzed. This position is always a polar basic (R/K/H). At the −2 UniProt numbering (1 within the Bowles/Koopman numbering), there was high variability in the amino acids used in SOX proteins (D/G/P/S/E), and only 86 ± 15% of the sequences were conserved with the human SOX member. In several of the SOX proteins, amino acids before this were conserved within the protein, but not across the family. On the other end of the HMG box, UniProt number 75 (77 of Bowles/Koopman numbering) could use a polar basic residue (R/K) in all, except for SOX30, and there was high conservation (93 ± 13%). At the next amino acid (76/78), there were seven different residues used in human SOX proteins (T/A/P/V/S/R/K) with only 87 ± 22% conservation. This would suggest that the best annotation of calling the overall HMG box was one amino acid shifted between the UniProt (−1 to 75) and the Bowles/Koopman numbering (2 to 77). 

The additive variants at each HMG box position showed an overlap of SOX proteins with high conservation for ClinVar-connected human clinical sequencing (Table 3). The −1 position was the first HMG box to have a variant, where SOX4 H58P was found to have uncertain significance in ClinVar (Accession VCV001526184.1). Eight amino acids of the HMG box had multiple variant changes at a site within one protein (Table 3, 9/11—SOX10; 16/18—SOX2, SOX9, SOX11, and SOX5; 29/31—SOX10; 32/34—SOX11, SOX5, and SOX10; 36/38—SOX10; 39/41—SOX10; 58/60—SOX10; and 72/74—SOX2 and SOX9). Twenty-seven amino acids of the HMG box had more than one SOX protein with a known ClinVar variant (Table 3, 1/3, 2/4, 3/5, 5/7, 6/8, 7/9, 9/11, 11/13, 12/14, 16/18, 19/21, 20/22, 28/30, 31/33, 32/34, 34/36, 36/38, 39/41, 47/49, 50/52, 54/56, 56/58, 57/59, 58/60, 61/63, 68/70, and 72/74).

The power of this ClinVar ortholog knowledge is in the ability to aid in the screening and interpretation of additional HMG box variants of SOX proteins. From these HMG box positions with a ClinVar variant and high conservation, we further assessed if any of the Geno2MP variants fell on one of these sites and if that individual had a matching phenotype or newly identified phenotype in more than one individual (Table 4). Of these 26 Geno2MP variants with a CADD score of >20 and selection score of ≥1, 6 variants had interesting phenotypes noted. SOX11 G84S (HMG box location 36/38) was found in one affected individual with eye abnormality with globe alteration, in addition to head or neck abnormality. SOX11 was associated with Coffin–Siris syndrome 9, commonly with dysmorphic facial features [16].

SOX14 R80Q (HMG box location 73/75) was found in two individuals with abnormality of the musculature, one with altered muscle physiology and another with muscular dystrophy. SOX14 did not have any OMIM annotated phenotypes. However, heterozygous global knockout of SOX14 has resulted in multiple altered mouse phenotypes, mostly morphological changes to various organs. but without any annotated changes to muscle phenotypes (https://www.mousephenotype.org/data/genes/MGI:98362, accessed on 18 December 2022). This suggests that the R to Q change may be conserved and that the phenotype is not connected to SOX14 variants within the individuals. SOX30 I367V (HMG box location 31/33) was absent from gnomAD and found in two individuals within Geno2MP with autism, intellectual disability, and mild cerebellar vermis hypoplasia. SOX30 variants have been associated with male infertility with testis-specific expression [80] and have never been associated with neurological traits. The conservation of the hydrophobic change suggests that this may not be the causal variant.

SOX15 R104G (HMG box location 56/58) was found in an individual with abnormality of the nervous system and musculature with noted fatigable weakness and progressive muscle weakness. Similar to SOX14, no current OMIM phenotypes have been annotated to SOX15, but SOX15 has been identified as critical for muscle differentiation and regeneration [81,82]. Position 56 was highly conserved as an R/K and found to have clinical variants in SOX2, SOX9, SOX5, and SOX6. The SOX 5 change was an R to G, matching that of the SOX15 Geno2MP variant, suggesting that it may be the individual’s causal pathogenic autosomal dominant variant.

SOX8 R159G (HMG box location 58/60) was absent in gnomAD and present in two individuals of Geno2MP with intellectual disability and microcephaly. SOX8 is not associated with any OMIM phenotypes, but it has been associated with developmental striatal projection neurons, glial cells, and the cerebellum [83,84]. The HMG box position 58 had ClinVar variants in the SOX10 homologous SOXE member linked to pathogenic annotation for Waardenburg syndrome type 2E, and SOX17 was also associated with disease due to changes at this site. None of the human SOX proteins utilized a G at this site, and R was highly selected in SOX8 evolutionary analysis, even with codon wobble occurring. This suggests that SOX8 R159G may be associated with altered neurological development. Thus, SOX15 with muscle disorders and SOX8 with neurological disorders may have supporting data within Geno2MP for novel genotype–phenotype associations.

### 3.4. HMG Box Variant SOX18 E137K

SOX18 E137K (HMG box location 53/55) was found within Geno2MP and gnomAD, while it was at a conserved site with known ClinVar variants in other SOX genes (Figure 5). The E137K of SOX18 fell on the third helix of the HMG box near the contacting sites with the first helix (Figure 5A). In the crystal structure of SOX18 (4y60), E137 formed a salt bridge with R140 and had hydrophobic packing with helix 1. E53 and R56 were highly conserved and selected in all 20 SOX proteins with multiple known ClinVar variants at several sites (yellow and magenta, Figure 5A). E137 was conserved in all 20 SOX genes in all 1890 sequences analyzed for the HMG box open reading frames. Over the HMG box sequences, the dN-dS value was −2.89. This amino acid was under greater than two standard deviations of selective pressure within SOX18, with a CADD score of 28.6 (near 0.1% of all deleterious variants) and a PolyPhen2 prediction of probably damaging. This suggests that E137K is of functional impact. Molecular dynamic simulations of the SOX18 protein showed that E137K resulted in elevated motion of the salt bridge (E/K137 with R140, https://youtu.be/CKg3dhkRHxY, accessed on 18 December 2022), which increased the availability of charges for potential protein interactions to increase, suggesting that it impacts SOX18’s function.

SOX18 E137K (rs201931544, 20_64048912_C_T) was found in 156 TOPMed sequenced humans and 274 gnomAD genomes. Within gnomAD, the variant was enriched in those with a Latino/Admixed American population background. The phase3 1000 genomes of subpopulations showed an allele frequency of 0.8% in MXL (Mexican Ancestry from LA, USA) with 24 additional SNPs in LD (r2 of 1), an allele frequency of 0.6% in PEL (Peruvians from Lima, Peru) with 44 SNPs in LD (r2 of 1), and an allele frequency of 0 in all other subpopulations. 

Rare variants in human SOX18 have been shown to result in Hypotrichosis–Lymphedema–Telangiectasia and renal syndrome (HLTRS), characterized by blood and lymphatic vascular symptoms and hair follicle defects [85]. In humans, HLTRS results from frameshift mutations that lead to the synthesis of a truncated version of the SOX18 protein, which acts as a dominant negative transcription factor, suppressing the endogenous functions of SOX7 and SOX17. In the case of SOX18 E137K, there was no frameshift in the ORF; however, it was not disregarded that amino acid variation in the third helix of the HMG-box interfered with protein partner recruitment. 

Within Geno2MP, SOX18 E137K was found in 11 affected individuals. Two individuals were annotated with abnormality of the head and neck with the sub-phenotype term for abnormality of the mouth. Multiple individuals had a neurological phenotype, including two individuals with seizures, one with microcephaly, and one with intellectual disability with autism. Two individuals had an abnormality of the cardiovascular system due to altered value development, one with abnormality of limbs and one with cloacal exstrophy (external abdominal organs). The number of neurological variants is interesting, as the homozygous knockout of mice results in significant abnormal behavior and decreased thigmotaxis (touch stimulus changes, https://www.mousephenotype.org/data/genes/MGI:103559, accessed on 18 December 2022). As there were no significant changes within heterozygous mouse knockouts of IMPC, and SOX18 loss of function variants were autosomal recessive for HLTRS [17], it suggests that SOX18 E137K may be autosomal recessive. 

Patients with SOX18 variants often suffer from capillary dysfunction, where mouse alterations of SOX18 do not modify the early vasculature in development, but rather the capillaries and lymphatics [86]. SOX18 has been shown to regulate the claudin-5 gene [87], which is critical for blood–brain barrier size selection [88]. The disruption of claudin-5 in mice results in spontaneous recurrent seizures and severe neuroinflammation [89]. Just recently, a patient with SOX18-associated HLTRS was observed to have idiosyncratic seizures following hyperbaric treatment [90]. This suggests that the neurological phenotypes of SOX18 E137K within Geno2MP may be environmentally regulated to contribute to changes in neuro-electrical control and, therefore, warrants additional characterization of functional outcomes for the missense variant.

SOX18 constructs were overexpressed in HeLa cells along with a SOX-regulated luciferase promoter, suggesting that the variant, E137K, decreased transcriptional regulation and was a loss-of-function (Figure 5B). SOX18 E137K additionally altered the interaction with known endothelial transcriptional regulators, such as MEF2C (adjP <0.0001) and RBPJ (adjP <0.0001), without disrupting dimerization to SOX7/17, as determined by the ALPHAScreen assay (Figure 5C), which measured pairwise protein–protein interactions. The analysis of a protein network built around a transcriptional regulatory node composed of SOX18, MEF2C, and RBPJ revealed a significant gene ontology (GO) enrichment for cardiovascular, circulatory, blood vessel, heart, and vasculature terms (Figure 5D). It should be noted that both MEF2C and RBPJ have been associated with intellectual disability and seizure disorders [91,92,93,94], matching the SOX18 E137K Geno2MP neurological phenotypes. This suggests a future need to study the role of SOX18 E137K in the blood–brain barrier and its role in neurological development and seizures.

### 3.5. Functional SOX Variants Outside the HMG Box 

In screening variants, it was noted that multiple variants fell within SOX proteins at conserved motifs or domains outside the HMG box. Known dimerization regions of several SOX genes, including SOX9 (N-term region before the HMG box) [95] and SOX18 (central region) [96], were conserved within the dataset (Figure 1). Multiple SOX genes had regions under high conservation/selection that have not been curated within databases (Figure 1, gray boxes). SOX4 had a conserved C-terminal sequence (SOX4 440-SGSHFEFPDYCTPEVSEMISG, underlined residues with conservation score ≥1) consisting of multiple aromatic and charged residues that predicted potential GSK3 and ProDKin kinase sites with multiple docking motifs for proteins such as CKS1 and MAPK. SOX5 had a linear motif (SOX5 110-SLSSTALGTPERRKGSLADVVDTLKQRKMEELIKNEPEETPS) with multiple conserved charged residues that made up multiple potential phosphorylation sites. SOX8 had two regions (SOX8 220- QTHGPPTPPTTPKTE and SOX8 255-GRQNIDFSNVDISELSSEVMGT), with the first having a degradation control switch with a sumoylation site and the second harboring potential phosphorylation sites. SOX10/11/12/14 had conserved C-terminal domains. This compiled vertebrate SOX analysis revealed multiple putative functional domains that warrant further detailed mechanistic studies.

We identified 56 variants found within ClinVar, Geno2MP, or gnomAD at highly conserved amino acids with additional amino acids around the site conserved, suggesting that the variant potentially impacted domains or motifs (Table 5). Several of these variants had matching phenotypes for the gene. SOX2 had two neurological-associated variants (D123G and G130A) found within the region flanking the HMG box with multiple conserved charges (HMGbox-PRRKTKTLMKK**D**KYTLPG**G**LLAPGGNSMA, bold underlined letters are sites of variants). SOX6 had several non-HMG box variants connected to neurological alterations, including H209K, S310T, R485Q, R545Q, and E591. As noted, the SOX9 N-terminal region for dimerization was conserved and had multiple Campomelic Dysplasia-linked variants within the motif (I73T, A76E, L81V, and G83R). Flanking the HMG domain of SOX10 were multiple variants from ClinVar (K179N, G181R, H216Q, and T240P). 

One of the most interesting variants outside of the HMG domain was that of SOX14 K88R (rs34393601). Within SOX14, the K was always conserved in all amino acid sequences of our alignment. The variant was predicted to be damaging in PolyPhen2 and had a CADD score of 25.8. The variant was found within 12 amino acids flanking the HMG box within a conserved motif with multiple prolines and charged residues (HMG box-PRRKPKNLL**K**KDRYVFPLPYLG). The most interesting observation is that, while the variant was present in gnomAD with an allele frequency of 0.018% of individuals (0.06% of north-western Europeans), it was present in five affected individuals with abnormality of the cardiovascular system, with an additional four with annotated thoracic aortic aneurysm. As noted above, SOX14 heterozygous knockout mice have been found to suffer from abnormal heart morphology (https://www.mousephenotype.org/data/genes/MGI:98362, accessed on 18 December 2022), suggesting some cardiovascular connections. 

Finally, SOX7 A379V was identified at the far C-terminal end of the protein, with high conservation of amino acids around it. SOX7/17/18 had annotated C-terminal conserved regions with additional high levels of conservation/selection at the last ~20 amino acids in each of the three proteins that have not previously been noted in the literature. Segregation analysis of SOX18 relative to SOX7/17 revealed multiple conserved amino acids, but could also elucidate several conserved amino acids in SOX18 that differed in SOX7 and SOX17 (Figure 6). The far C-terminal region contained multiple highly conserved serine and threonine amino acids with the potential for phosphorylation and multiple hydrophobic amino acids, making this an ideal protein-interacting peptide. Moreover, the identification of ClinVar, Geno2MP, and gnomAD variants within this region suggests a need for SOX gene variant insights outside the HMG box. 

SOX7 A379V (rs143587868) was found to be heterozygous in 0.716% of African individuals from gnomAD, as confirmed in the TOPmed Bravo dataset (allele frequency of 0.2% total), and had a CADD score of 29.0 (near 0.1% of all deleterious variants). The variant was found on the most highly conserved motif of SOX7 (Figure 1), near multiple conserved Ser/Thr sites with potential phosphorylation, and was conserved on the C-terminal ends of SOX7, SOX17, and SOX18. We speculate that this region is critically involved in the phosphorylation control of SOXF genes; however, there is no information regarding the function of post-translational modification of these TFs. Within Geno2MP, SOX7 A379V was found in six affected individuals, with four having abnormal eye phenotypes (glaucoma or retinal degeneration). Glaucoma occurs five times more often in African American individuals [97], in which this variant is enriched. The SOXF genes have been identified to regulate the vascular development of the eye [98]. SOX7 A379V represents a novel potential for the future characterization of the C-terminal ends of SOXF genes and their roles in eye phenotypes.

### 3.6. Somatic SOX Variants in Cancer

Our final genomics analysis of SOX proteins was the analysis of somatic variants within cancer using the COSMIC database. As of December 2022, there were 4313 SOX-based COSMIC variants. Similar to ClinVar, where variants were enriched within the HMG box of several proteins, cancer somatic variants also showed an HMG box enrichment for SOX11, SOX9, SOX4, SOX10, SOX1, SRY, SOX18, SOX21, SOX3, SOX12, SOX14, SOX17, SOX2, and SOX13 (cyan, Figure 7A). Nearly all of the SOX proteins had a significant de-enrichment for variants within the HMG box for gnomAD and Geno2MP. This suggests that functional SOX somatic variants might be elevated within cancer samples. Most of the SOX HMG box variants fell on conserved sites (Figure 7B), with multiple sites of the HMG domain annotated for ClinVar or the literature’s phenotype connections (Figure 7C). These HMG box variants were primarily found in Adenocarcinoma and large intestine samples (Figure 7D, 117 samples), with many of the SOX proteins having high-risk variants for the pathology. Of the top-ten cancer types, SOX9 had the leading high-risk variants for large intestine adenocarcinoma and stomach non-specified (NS) cancer. SOX17 was the leading protein for skin NS cancer, stomach adenocarcinoma, endometrium endometrioid carcinoma, lung adenocarcinoma, and urinary tract NS cancer. SOX13 accounted for 22% of the high-risk variants in lung squamous cell carcinoma, SOX 2 for 33% of pancreas ductal carcinoma, and SOX7 for 40% of thyroid NS cancer. This suggests that further analysis of somatic variants within SOX proteins may be critical and that the gene family has functionality outside early development.

## 4. Discussion

Since 1990, with the discovery of *SRY* variants in sex reversal [57], *SOX* genes have had known disease implications [18,99]. Our group characterized the rat chromosome-Y *Sry* duplications and point mutations involved in blood pressure regulation [41,100]. One of the duplicated *Sry* genes, *Sry2*, found on the rat Y-chromosome, was inserted into the intron of *Kdm5d*, driving a ubiquitous expression profile [41]. We hypothesized that this ubiquitous expression profile of the *Sry2* gene altered the phenotype of an ancient rat, such that a loss-of-function mutation (R21H, HMG box location 17) occurred and was selected to compensate for the ubiquitous expression of *Sry2*. This insight led us to explore additional *SRY* and *SOX* gene variants that might impact phenotypes while attempting to understand the molecular mechanisms of the variants. While this initially appeared to be an easy task, we were surprised to find a lack of functional domain/motif mapping in *SOX* genes outside the HMG box. This has recently been identified as an emerging need within SOX gene knowledge [18]. 

Therefore, we developed a deep evolutionary assessment of *SOX* ORF sequences to map functional sites (Figure 1), followed by an assessment of human variants in gnomAD, COSMIC, the published literature, and ClinVar. Future work could be focused outside vertebrate *SOX* genes, focusing on comparative evolutionary analysis of invertebrate species, which was not performed here. Further work on domain annotations could also be performed. Conservation analysis can be challenging [101]. Utilizing the codon selection strategy described in this work [22,23], a few divergent sequences decreased their conservation scores. For example, the loss of conservation in several gene members shown in Figure 1 was likely the result of a few divergent species sequences. This is why amino acid alignments were performed with more species, yielding insights over all species that were not biased by a few divergent sequences.

Before the 2000s, most variant insights were published in the literature [102], and therefore we developed a list of common SOX genetic variant papers in Table 1. This table is not meant to be all-inclusive, but captures the most notable SOX genetic variant papers. Following the establishment of ClinVar and other databases, it became more typical for variants to be listed in both publications and within the database. Thus, combining the early variant manuscripts (Table 1) with the most common variant databases makes a larger map of the genomics landscape of SOX genes possible.

Using multiple data sources, this study has reaffirmed many published observations while making multiple novel discoveries: (1) uncharacterized, but highly conserved, motifs found in *SOX2*, *SOX14*, *SOX4*, *SOX11*, *SOX12*, *SOX5*, *SOX8*, and *SOX10*; (2) 100% conservation of all HMG box DNA-specific contact amino acids; (3) enrichment of variants in the HMG boxes of multiple SOX proteins; (4) potential high-impact human variants in SOX18 and SOX7; (5) potential novel syndromes for SOX15 in musculature abnormalities and SOX8 with intellectual disabilities; and (6) the role of functional variants outside the HMG box, including those of SOX2, SOX9, SOX14, and SOX7. 

In our COSMIC analysis, we can confirm the recent findings of the Jauch lab [19] that a few cancer variants are found at critical sites of the SOX2/17-OCT4 interaction residues [9,103] (HMG boxes numbered 44, 48, 51, 55, 62, and 69). Most notable is the role of amino acid changes to SOX9 at position 62 (deletion of residue K), seen multiple times in cancer. These amino acids represent one of the fascinating parts of the evolution of *SOX* genes, specifically HMG box sites 44, 55, and 62 (Bowles annotated sites 46, 57, and 64), where our evolutionary conservation shows high intra-protein multiple-species conservation while variation across SOX proteins.

Understanding the clinical impact of variants between 0.01 and 1% is incredibly challenging [23]. This can be seen for our identified SOX18 and SOX7 human variants. SOX18 E137K (rs201931544) was not found on most SNP-Chips used in genotyping patients, thus making phenotypic association rather challenging. This indicates that the variant and all of the LD SNPs of Latino individuals have been included in limited published genome-wide association studies (GWAS). Genotyped individuals in large biobanks cannot have phenome-wide association studies (PheWAS) performed for this site due to the low allele frequency, underpowering discovery. The lower limits of detectable allele frequencies for significance within GWAS and PheWAS are greater than 1–5% of the global population [104], while SOX18 E137K is estimated to have a frequency between 0.1% and 0.05% globally. This highlights the ethnic/racial bias currently present in genomic knowledge that needs to be actively addressed [105]. 

To compensate for this lack of information, we took an alternative approach based on a preliminary molecular assessment of the functional impact of SOX18 E137K. This revealed a change in transcriptional activity and affinity for protein partners, likely yielding a change in transcription factor biology. Our preliminary laboratory experiments on SOX18 E137K suggest a loss of function by the variant. Additional molecular and biochemical experiments, including co-immunoprecipitations, nuclear localization, DNA binding, and structural changes, are required to further elucidate the changes to SOX18 biology. Most notably, based on the Geno2MP annotated phenotypes, we suggest that it is critical to characterize the gene within the endothelial cells of the blood–brain barrier. As whole-genome sequencing is being implemented in more studies, particularly for Latino populations, the variant will be identified more often and could be linked to phenotypes. This highlights the increasing importance of extensive data integrations into genomic medicine, an undertaking within the All of Us precision medicine initiative [106] that will make the systematic assessment of gene variants more interpretable. 

As sequencing increases in multiple species and millions of humans, systematic assessments of gene families are needed. This paper uses bioinformatics to assess the genotype-to-phenotype associations of each of the 20 human *SOX* gene family members. This work highlights the utility of studying gene families with a more systematic approach, which should be applied to more transcription factor families to segregate genotype-to-phenotype associations further, which is not possible when studying only a single gene family member. These strategies will ultimately help when prioritizing understudied, yet fruitful, avenues of genetic research within gene families in the age of data integration.

## 5. Conclusions

This study is the first to systematically analyze the evolutionary conservation within thousands of sequences of SOX genes/proteins with multiple database integrations of human variants linked to phenotypes. From discovering novel domains outside the HMG box, linking several SOX genes to novel phenotypes, and identifying several inherited variants linked to phenotypic traits, we show the promise of new genomic discoveries within a large transcription factor family. While we continue to advance our knowledge of SOX members, this work also highlights the importance of using paralog mapping to understand variants better, especially when occurring in a paralog family member that is less studied with new knowledge to be gained. Overall, this shows that, even in 2023, there is much genomic knowledge to be learned, and bioinformatics holds much promise in advancing our genomic insights.

## Figures and Tables

**Figure 1 genes-14-00222-f001:**
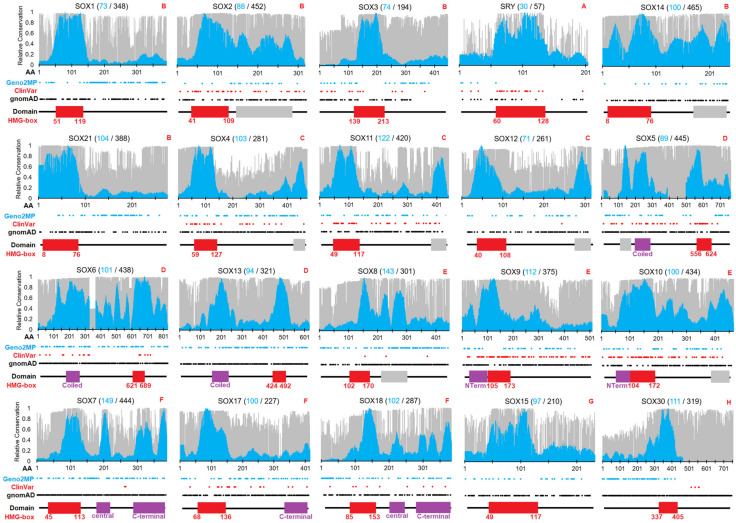
Conservation and variants for each of the 20 SOX genes/proteins. In total, 1890 nucleic acid sequences (cyan) and 6667 amino acid sequences (gray) were analyzed for 20 SOX genes throughout vertebrate species. The evolutionary selection was mapped for each gene using the indicated number of sequences on a scale of 0 (weak conservation) to 1 (high conservation). Shown below the conservation data are the locations of human variants from Geno2MP (cyan), ClinVar (red), and gnomAD (black). UniProt-designated domains are shown below, each with the HMG box identified in red. HMG box human numbers are shown below the domain. Red letters at top right corner indicate the SOX subfamily annotation. The raw TIF file can be found at https://doi.org/10.6084/m9.figshare.21830343.v1 (accessed on 18 December 2022).

**Figure 2 genes-14-00222-f002:**
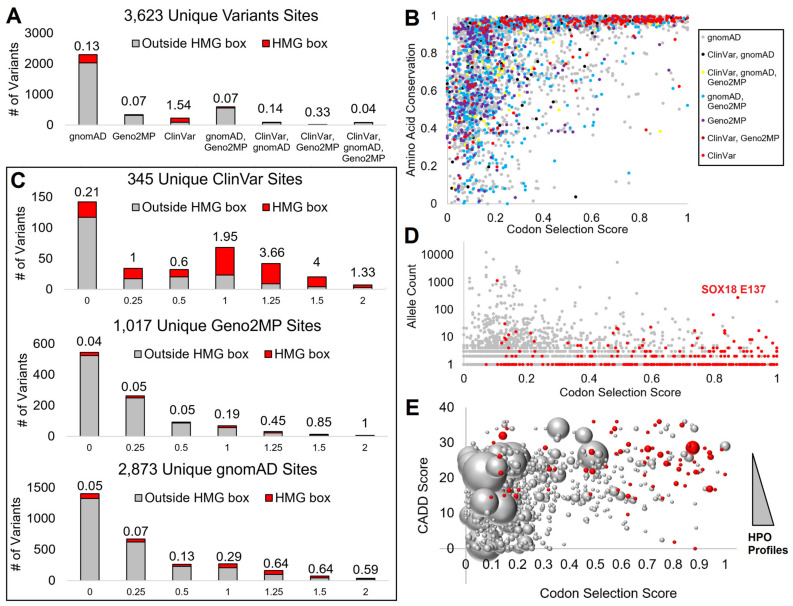
Mapping SOX variants to conserved amino acids. (**A**) Assessment of each amino acid position and the number of variants at that site from each database. Those within the HMG box are in red and those outside the HMG box are in gray, with the value listed above each showing the ratio of variants within the HMG domain. If a variant, whether with the same or a different change, is found in multiple genomics databases for an amino acid, it is listed under the combination of databases. (**B**) Codon selection score (x-axis) relative to the amino acid conservation (y-axis) for each amino acid, colored as grouped in panel (**A**). (**C**) Codon selection score of unique sites for ClinVar, Geno2MP, and gnomAD, where the higher values indicate codon selection. Red represents variants within the HMG box and gray represents variants outside the HMG box. The number above each is the ratio of HMG to non-HMG variants. (**D**) Codon selection score (x-axis) relative to allele count (y-axis) of gnomAD variants. Those in red are within the HMG box. (**E**) Codon selection score (x-axis) relative to the functional variant CADD score of the highest annotation (y-axis). HMG box variants are in red and the bubble size represents the number of human phenotype profiles (HPO).

**Figure 3 genes-14-00222-f003:**
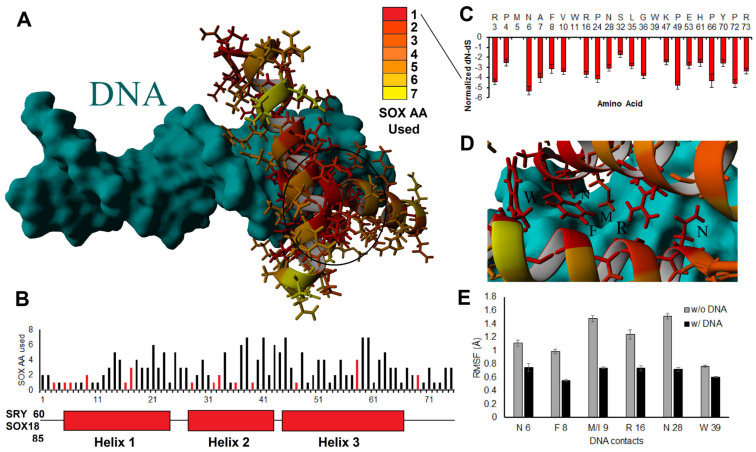
SOX genes and conservation of DNA contacts. (**A**,**B**) Usage of different amino acids throughout the 1890 sequences shown on the structure of the HMG box (**A**) or the three-helix pictorial (**B**). In panel (**A**), we have circled the area of high conservation that drives the HMG box three-helix structure. In panel (**B**), we have included the amino acid in SRY or SOX18 that correspond to HMG box amino acid 1. The bars in red are amino acids that are in contact with the DNA. (**C**) The amino acids that are 100% conserved in all sequences analyzed show the relative selection at each site throughout evolution using a dN-dS metric. A dN-dS metric is the rate of nonsynonymous variants throughout evolution relative to synonymous variants. The more negative this value is, the more selective pressure there is to maintain the amino acid based on codon wobble. (**D**) Location of the 100% conserved sites relative to DNA binding on the structure. (**E**) Movement (root-mean-squared fluctuation, RMSF, in Å) of the six critical DNA contacts of SOX HMG proteins labeled in panel (**D**) following 20 nanoseconds of molecular dynamic simulations without DNA (gray) and with DNA (black) for all 20 structures of the SOX HMG domains. Error bars represent the standard error of the mean.

**Figure 4 genes-14-00222-f004:**
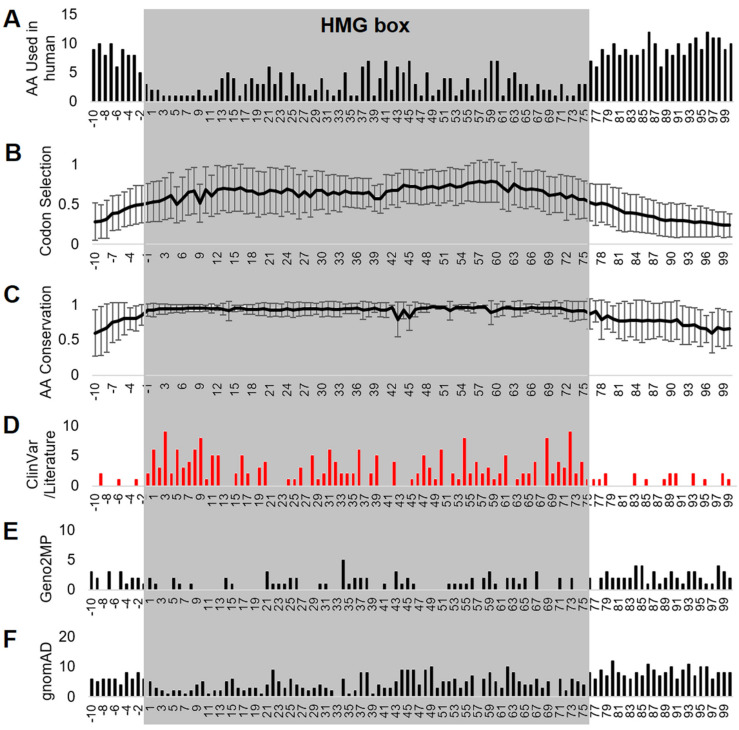
HMG box alignment of all SOX proteins. Alignment of all HMG box sequences of the 20 SOX proteins showing the number of human amino acids used (**A**), codon selection scores (**B**), conservation (**C**), and genomic variants for different SOX proteins using ClinVar or the literature (**D**), Geno2MP (**E**), or gnomAD (**F**). Error bars for panels (**B**,**C**) represent plus and minus the standard error of the mean over all 20 genes/proteins. Amino acid 1 is the V/I annotated within UniProt as the first amino acid of the HMG box for each protein. Alignments were performed without any gaps.

**Figure 5 genes-14-00222-f005:**
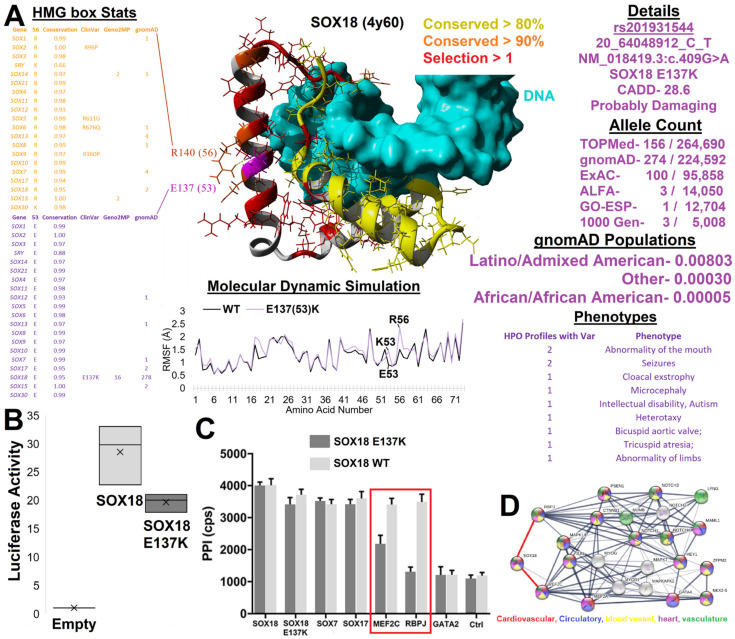
SOX18 E137K. (**A**) Integrated data and structure of SOX18 E137K. In the middle is the SOX18 structure (PDB file 4y60) bound to DNA (cyan). Amino acids are colored based on conservation and selection as labeled. To the left of the structure is the annotation of the HMG box compiled data for R56 (yellow) and E53 (magenta) amino acids. Below are 20 nanoseconds of mds for wild type (WT, black) or E137K (magenta), showing the root-mean-squared fluctuation for each amino acid. To the right are various data insights for E137K, including the multiple annotations, allele counts, and Geno2MP phenotypes. (**B**) Transcriptional activity of empty plasmid (empty normalized to 1), with SOX18 (gray) or SOX18 E137K (black) overexpressed in combination with an SOX response element driving luciferase production in HeLa cells. (**C**) ALPHAScreen assay to assess pairwise interactions between SOX18 WT or the SOX18 E137K variant with known protein partners. GATA2 is a known nonbinding control that provides a baseline ALPHAScreen signal similar to the control condition with buffer only (Ctrl). Error bars represent the standard error of the mean of three independent experiments. (**D**) STRING analysis of SOX18, MEF2C, and RBPJ with the first and second shells of the network with no more than ten interactions added in. Colors represent GO enrichments as follows: red—cardiovascular system development (GO:0072358; FDR—9.18 × 10^−15^); blue—circulatory system development (GO:0072359; FDR = 9.18 × 10^−15^); yellow—blood vessel development (GO:0001568; FDR = 2.67 × 10^−13^); magenta—heart development (GO:0007507; FDR = 2.67 × 10^−13^); green—vascular development (GO:0001944; FDR = 3.55 × 10^−13^). The raw TIF file can be found at https://doi.org/10.6084/m9.figshare.21830421.v1 (accessed on 18 December 2022).

**Figure 6 genes-14-00222-f006:**
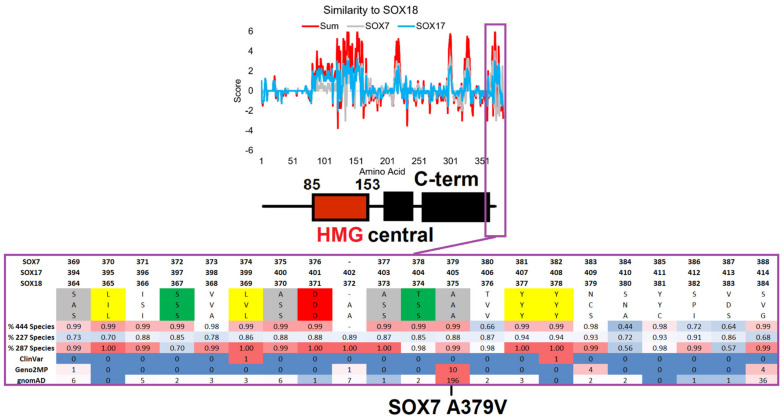
Conservation of the SOX-F proteins highlighting the far C-terminal region. The top panel shows a comparative analysis of SOX18 relative to the SOX7 (gray) and SOX17 (cyan) conservation scores for each amino acid of SOX18, with the SOX18 domain annotation from Figure 1 shown below. When the amino acids are the same between SOX18 and the others, they have positive values; when they are different, the values are negative. The red line is the additive value of the SOX7 and SOX17 comparisons. In the bottom panel, the magenta call out shows the conservation of amino acids within the far C-terminal region of SOX-F members. The amino acid numbers of each protein are provided on top of the sequences. This is followed by the human amino acids, where those highlighted are conserved in each sequence (gray—flexible; yellow—hydrophobic; green—S or T; red—polar acidic). Then, the percentage of conservation for the amino acid sequence alignments is shown, with a value of 1 indicating 100% conservation. At the bottom is the number of variants seen in each genomics database. The conservation and variants are shown on a heatmap, with red representing the highest values and blue representing the lowest.

**Figure 7 genes-14-00222-f007:**
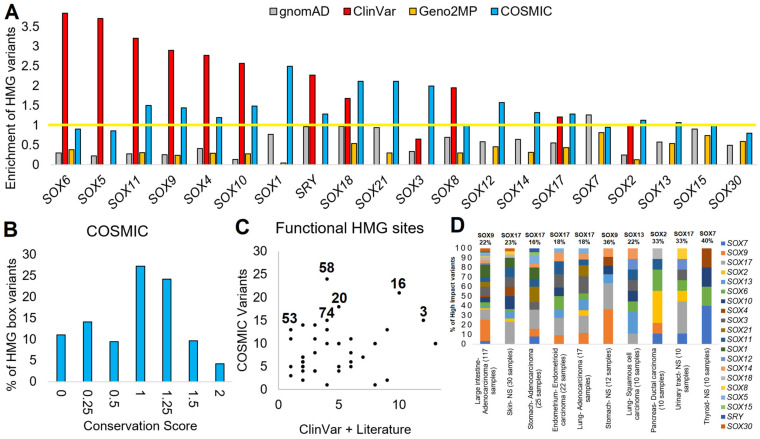
COSMIC variants of SOX genes. (**A**) Enrichment of variants within the HMG box for each SOX protein for gnomAD (gray), ClinVar (red), Geno2MP (orange), or COSMIC (cyan). Genes are ranked based on the highest enrichment for any database. The enrichment was calculated by normalizing the HMG box size relative to the total protein such that a value of 1 (yellow line) is the random probability of a variant falling within the HMG box. (**B**) Codon selection score of variants from COSMIC that fall with the HMG domain. (**C**) Amino acids found with ClinVar or annotations in the literature (x-axis) associated with disease for the HMG box that have COSMIC variants (y-axis) (**D**) Annotation of HMG box variants from panel (**C**) showing tissue type and histology annotations for each protein of the top-10 tumor types. The number of COSMIC samples with each cancer type with high-impact variants is shown on the x-axis labels. The top SOX gene for the percent of variants is labeled on top of each bar.

**Table 3 genes-14-00222-t003:** ClinVar variants within the HMG box using UniProt or Bowles/Koopman numbering. Rotating white to gray indicates different amino acids with multiple variants for some.

UniProt	Bowles/Koopman	AA Used	AA Used	AA Conservation	Publications	Unique ClinVar	gnomAD Count	HPO Profiles	Variant	Syn (s)	Nonsyn (n)	Conservation Score
−1	2	3	RHK	0.93 ± 0.08		2	6	1	**SOX4 H58P**	4	0	1
1	3	2	VI	0.92 ± 0.09	3	3	6	3	**SOX4 I59S**	7	0	1.25
									**SOX9 V105F**	11	0	1
2	4	2	KR	0.94 ± 0.07		3	3	1	**SOX3 K140R**	10	0	2
									**SOX9 K106E**	9	0	1.25
									**SOX10 K105Q**	4	0	1
3	5	1	R	0.94 ± 0.07	2	10	2	0	**SOX4 R61Q**	22	0	1.5
									**SOX5 R558H**	10	0	1.25
									**SOX6 R623Q**	9	0	1.25
									**SOX10 R106G**	15	0	1.25
5	7	1	M	0.95 ± 0.05	2	4	21	2	**SOX6 M625T**	0	0	1
									**SOX10 M108T**	0	0	1
6	8	1	N	0.95 ± 0.05		3	1152	1	**SOX5 N561H**	10	0	1.5
									**SOX10 N109S**	16	0	2
7	9	1	A	0.95 ± 0.05		4	2	0	**SRY A66P**	6	0	1.5
									**SOX4 A65T**	17	0	1.25
									**SOX9 A111T**	13	0	1
									**SOX10 A110V**	2	0	1
9	11	2	MI	0.95 ± 0.04	4	9	4	0	**SOX9 M113V/I**	0	0	1
									**SOX10 M112V/R/T/I**	0	0	1
10	12	1	V	0.95 ± 0.04		1	5	0	**SOX9 V114L**	12	0	1
11	13	1	W	0.95 ± 0.06	1	4	1	0	**SOX11 W59R**	0	0	1
									**SOX9 W115R**	0	0	1
									**SOX6 W631C**	0	0	1
12	14	2	SA	0.96 ± 0.03		5	2	0	**SOX11 S60P**	20	0	1.5
									**SOX9 A116V**	23	0	1.5
15	17	4	QEAH	0.95 ± 0.04	1	1	29	2	**SOX9 A119E**	15	0	1.25
16	18	1	R	0.95 ± 0.04		10	4	0	**SOX2 R56G/W**	13	0	1.25
									**SOX9 R120G/L**	20	0	1.25
									**SOX11 R64C/G/H**	16	0	1.25
									**SOX10 R119L**	8	0	1
									**SOX5 R571L/W**	16	0	1.25
19	21	3	MIL	0.93 ± 0.08	1	2	6	0	**SOX4 I77V**	12	0	1.5
									**SOX10 L122V**	20	0	1.25
20	22	3	AML	0.94 ± 0.09	1	4	9	0	**SOX9 A124P**	35	1	1
									**SOX10 A123P**	15	0	1.25
24	26	1	P	0.94 ± 0.10		1	14	6	**SRY P83H**	2	0	1.25
26	28	3	MLA	0.94 ± 0.09		3	10	2	**SOX10 L129P**	11	0	1
28	30	1	N	0.94 ± 0.09	2	3	16	0	**SOX11 N76D**	4	0	1
									**SOX5 N583S**	9	0	1.25
									**SOX17 N95S**	9	0	1.5
29	31	2	SA	0.94 ± 0.08		2	3	0	**SOX10 A132G/V**	19	0	1.5
31	33	2	IL	0.94 ± 0.10	3	3	4	2	**SOX11 I79L**	12	0	1.5
									**SOX10 L134P**	4	0	1
32	34	1	S	0.94 ± 0.11	1	7	3	0	**SOX11 S80F/C**	2	0	1
									**SOX5 S587C**	1	0	1
									**SOX10 S135R/G/T**	5	0	1
33	35	2	KV	0.93 ± 0.10	1	1	0	0	**SOX5 K588N**	5	0	1
34	36	5	RQITM	0.95 ± 0.06		2	21	9	**SOX2 R74P**	16	0	1.5
									**SOX9 T138K**	27	0	1.5
35	37	1	L	0.95 ± 0.07	1	1	1	1	**SOX4 L93Q**	11	0	1
36	38	1	G	0.95 ± 0.07	2	6	3	2	**SOX2 G76D**	8	0	1.25
									**SRY G95R/E**	2	0	1.25
									**SOX9 G140D**	14	0	1
									**SOX10 G139C/D**	10	0	1
38	40	7	EDQRLSA	0.93 ± 0.10		2	23	3	**SOX10 L141P**	11	0	1
39	41	1	W	0.96 ± 0.06	1	7	1	0	**SOX4 W97G**	0	0	1
									**SOX6 W659R**	0	0	1
									**SOX10 W142R/S/C**	0	0	1
47	49	1	K	0.96 ± 0.06	2	3	24	0	**SRY K106I**	1	0	1.25
									**SOX10 K150E**	5	0	1
									**SOX4 K105N**	1	0	1
48	50	5	RWIQK	0.96 ± 0.05	1	2	12	0	**SOX10 R151P**	15	0	1.25
50	52	2	FY	0.96 ± 0.02	2	5	11	0	**SRY F109S**	2	0	1.25
									**SOX9 F154L**	14	0	1.5
									**SOX10 F153I**	13	0	1.5
52	54	4	DQRE	0.92 ± 0.13		2	11	1	**SOX10 E155K**	6	0	1
53	55	1	E	0.97 ± 0.02		1	285	16	**SOX18 E137K**	10	0	2
54	56	2	AQ	0.97 ± 0.03	3	6	9	1	**SRY A113T**	5	0	1.5
									**SOX11 A102V**	16	0	1.25
									**SOX10 A157V**	28	0	2
									**SOX4 A112P**	11	0	1.25
55	57	4	KQEA	0.96 ± 0.04		2	9	1	**SOX17 E122D**	3	0	1
56	58	2	RK	0.96 ± 0.07		4	14	4	**SOX2 R96P**	6	0	1
									**SOX9 R160P**	19	0	1.25
									**SOX5 R611G**	22	0	1.5
									**SOX6 R676Q**	13	0	1.25
57	59	2	LI	0.97 ± 0.01		2	0	0	**SOX2 L97P**	5	0	1
									**SOX10 L160P**	12	0	1
58	60	4	RQSK	0.97 ± 0.02	1	3	76	2	**SOX10 R161C/H**	15	0	1.25
									**SOX17 R125S**	18	0	1.25
61	63	1	H	0.95 ± 0.10	2	4	3	0	**SOX2 H101R**	1	0	1
									**SOX9 H165Y/R**	7	0	1.25
									**SOX10 H164P**	5	0	1
63	65	5	KEAQR	0.94 ± 0.10		1	26	2	**SOX10 K166E**	11	0	1.25
65	67	3	HYF	0.95 ± 0.06		2	4	3	**SOX11 Y113C**	16	1	1
68	70	2	YW	0.95 ± 0.08	3	6	4	0	**SOX11 Y116C**	3	0	1
									**SOX17 Y135C**	3	0	1
70	72	1	Y	0.95 ± 0.11		4	0	0	**SOX4 Y128H**	2	0	1
71	73	3	RKQ	0.95 ± 0.08	1	3	11	4	**SOX10 Q174P**	8	0	1.25
72	74	1	P	0.93 ± 0.13	4	9	4	0	**SOX2 P112T/A**	11	0	1.25
									**SOX11 P120L**	13	0	1.25
									**SOX9 P176T/S/L/R**	35	1	1
									**SOX6 P692S**	11	0	1.25
									**SOX10 P175S**	17	0	1.25
73	75	1	R	0.91 ± 0.14		3	25	4	**SOX2 R113W**	4	0	1
74	76	3	RKP	0.92 ± 0.13	1	3	6	0	**SOX10 R177Q**	25	0	1.5

**Table 4 genes-14-00222-t004:** Geno2MP variants within the HMG box using UniProt numbering. Those in red have matching phenotypes to the gene OMIM or in multiple Geno2MP individuals. Rotating white to gray indicates different SOX genes with multiple variants for some.

Gene	Geno2MP Var	HMG Box #	AA Used in Human	AA Conservation	Genes with ClinVar	Phenotype	Geno2MP HPO	Geno2MP CADD	gnomAD Count	Syn (s)	Nonsyn (n)	Conservation Score
*SOX1*	E88A	38	7	0.93 ± 0.10	2	Microcephaly	1	21.6	4	5	0	1.5
*SOX10*	H128Q	25	5	0.93 ± 0.10	1	Abnormality of the ear	1	23.8	0	4	0	1
*SOX10*	L138P	35	1	0.95 ± 0.07	1	Nephrotic syndrome	1	21.2	0	11	0	1
* SOX11 *	G84S	36	1	0.95 ± 0.07	4	Abnormality of the globe	1	28.5	0	13	0	1.25
*SOX13*	Q444E	21	6	0.92 ± 0.09	0	Abnormality of the limb	2	28.6	1	16.5	2.5	1
*SOX13*	R461H	38	7	0.93 ± 0.10	2	Hypoplastic left-heart	2	34	8	17	0	1.25
*SOX13*	A478E	55	4	0.96 ± 0.04	2	Retinitis pigmentosa	1	28.5	2	24	0	1.5
*SOX14*	R63Q	56	2	0.96 ± 0.07	4	Congenital diaphragmatic hernia	2	29.2	1	19	1	1
* SOX14 *	R80Q	73	1	0.91 ± 0.14	2	Abnormal muscle physiology (×2)	3	26	0	11	0	1.25
*SOX15*	R82L	34	5	0.95 ± 0.06	2	Retinitis pigmentosa	2	36	9	7	0	1.25
*SOX15*	G84D	36	1	0.95 ± 0.07	4	Tricuspid atresia	1	33	2	11	0	1.25
* SOX15 *	R104G	56	2	0.96 ± 0.07	4	Progressive muscle weakness	2	27.2	0	14	0	1.25
*SOX17*	M101V	34	5	0.95 ± 0.06	2	Distal arthrogryposis	1	23.2	0	0	0	1
*SOX17*	R125S	58	4	0.97 ± 0.02	2	Neural Atrophy/Degeneration	1	33	4	18	0	1.25
* SOX18 *	E137K	53	1	0.97 ± 0.02	1	Seizures (×2), DD/ID, ASD, brain morphology, heart	16	28.6	278	10	0	2
*SOX18*	N151S	67	3	0.95 ± 0.05	0	abdominal organs	1	24	1	3	0	1
*SOX18*	R155Q	71	3	0.95 ± 0.08	2	Malformation of the heart (×2)	3	26.6	1	15	0	1.25
* SOX30 *	I367V	31	2	0.94 ± 0.10	3	ASD, Cerebellar hypoplasia (×2)	2	22.5	0	3	0	1
*SOX30*	E399A	63	5	0.94 ± 0.10	1	cardiovascular system	1	24.5	3	7	0	1.5
*SOX6*	D607E	14	5	0.91 ± 0.13	0	Abnormality of the cerebral cortex	1	25.1	0	16	0	2
*SOX6*	M619T	26	3	0.94 ± 0.09	3	Thoracic aortic aneurysm	1	21.5	0	0	0	1
*SOX7*	K81R	37	6	0.92 ± 0.12	0	Spontaneous abortion	2	22.2	1	12	0	1.25
*SOX7*	A98G	54	2	0.97 ± 0.03	5	Abnormality of hindbrain morphology	1	27.8	1	33	0	1.5
*SOX7*	D108N	64	3	0.95 ± 0.11	2	central nervous system	2	35	3	17	0	1.5
* SOX8 *	R159G	58	4	0.97 ± 0.02	2	Intellectual disability, microcephaly (×2)	1	20.8	1	40	0	1.5
*SOX8*	K163R	62	4	0.96 ± 0.02	0	Nephrotic syndrome	1	35	5	10	0	1.25

**Table 5 genes-14-00222-t005:** High-ranking SOX protein variants found outside the HMG box. Red text are those that have matching phenoptypes or are of high interest. Rotating white to gray indicates different SOX genes with multiple variants for some.

Gene	Codon #	AA	Var	Phenotype	Syn (s)	Nonsyn (n)	Selection Score	21 Codon Window	AA Conservation	gnomAD Count	Geno2MP	HPO Profiles	CADD
*SOX1*	126	T	T126I	Abnormality of hindbrain morphology	1	0	1	25.25	0.97	1	T126I	1	21.7
* SOX2 *	123	D	D123G	Developmental disorder	4	0	1.25	17.5	1.00	1			
* SOX2 *	130	G	G130A	abnormalities of the central nervous system	8	0	1.25	15.25	0.99	17	G130A	1	23.3
*SOX2*	133	A	A133T	Anophthalmia/microphthalmia-esophageal atresia syndrome	22	0	2	11.25	0.96	8	A133T	3	23.3
*SOX2*	272	D	D272N	not provided	1	0	1	13.75	0.97				
*SOX5*	135	R	R122H	Intellectual disability	10	0	1.25	23.75	0.94	1	R122H	1	35
*SOX5*	159	P	P146L	Thoracic aortic aneurysm	10	0	1.25	14.75	0.94	1	P146L	1	25.1
*SOX5*	206	I	I206V	Lamb-Shaffer syndrome	4	0	1	19.75	0.95				
*SOX5*	228	A	A215V	Coarctation of aorta	22	0	1.5	20.5	0.95	10	A215V	1	33
*SOX5*	261	K	K261N	not provided	4	0	1	22.5	0.96				
*SOX5*	266	Q	Q266H	Lamb-Shaffer syndrome	7	0	1.25	16.25	0.96				
*SOX5*	268	Q	Q268H	Lamb-Shaffer syndrome	5	0	1	14.5	0.96				
*SOX6*	146	E	E146K	not provided	10	0	1.5	13	0.95	2	E146D	2	20.4
* SOX6 *	209	H	H209K	Multiple neurological	16	0	2	22.25	0.97				
*SOX6*	214	K	K214Q	Hypoplastic left-heart syndrome	8	0	1.25	23.5	0.97	4	K214Q	2	22
*SOX6*	252	N	N252K	not provided	3	0	1	18.25	0.97				
*SOX6*	264	M	M264I	myopathy	0	0	1	17.25	0.96	5	M264I	1	22.1
*SOX6*	277	R	R277W	not provided	7	0	1	19.75	0.97	14			
*SOX6*	280	A	A280E	Aplasia/Hypoplasia affecting the eye	4	0	1	18.75	0.97	1	A280E	2	23.1
*SOX6*	281	A	A281T	not provided	8	0	1.25	19	0.97	1			
*SOX6*	291	F	F291L	cardiovascular system	7	0	1.25	17	0.97	5	F291L	1	22.4
* SOX6 *	310	S	S310T	Microcephaly	5	0	1	13.25	0.97	56	S310T	2	21.6
*SOX6*	312	M	M312V	not provided	0	0	1	13.75	0.95				
*SOX6*	371	A	A330P	Aplasia/Hypoplasia affecting the eye	10	0	1.25	14	0.99	1	A330P	1	23.9
* SOX6 *	512	R	R485Q	Abnormality of nervous system	12	0	1.25	17	0.99	2	R485Q	1	28.5
* SOX6 *	572	R	R545Q	Intellectual disability	9	0	1.25	15	0.99	6	R545Q	1	32
* SOX6 *	618	E	E591K	Intellectual disability	10	0	1.5	20.25	0.98	1	E591K	1	35
*SOX7*	128	R	R128C	abnormality of the central nervous system	30	0	1.5	12	0.99	4	R128C	1	21.8
*SOX7*	329	R	R329H	Nephrotic syndrome	19	0	1.25	15.25	1.00	3	R329H	1	25.3
* SOX7 *	379	A	A379V	Abnormality of the eye (×4)	25	0	1.25	21	0.99	195	A379V	9	29
*SOX8*	263	N	N263I	Abnormality of the nervous system (×2)	17	0	1.25	13.25	0.99	45	N263I	6	24.1
* SOX9 *	73	I	I73T	Campomelic dysplasia	2	0	1	14.25	0.97				
* SOX9 *	76	A	A76E	Inborn genetic diseases	32	0	2	17	0.97				
* SOX9 *	81	L	L81V	Campomelic dysplasia	20	0	1.25	18.25	0.97	1			
* SOX9 *	83	G	G83R	Inborn genetic diseases	10	0	1	20.25	0.97				
* SOX10 *	68	F	F68L	PCWH syndrome	4	0	1	12.5	0.96	1			
*SOX10*	75	A	A75V	Malformation of the heart and great vessels (×2)	20	0	1.5	17.25	0.96		A75V	2	34
*SOX10*	92	V	V92M/L	PCWH syndrome	7	0	1	17.5	0.97	46	V92L	4	24.1
* SOX10 *	179	K	K179N	not provided	3	0	1	19.25	1.00				
* SOX10 *	181	G	G181R	not provided	9	0	1	17.5	0.98	1			
* SOX10 *	216	H	H216Q	not provided	4	0	1	13	0.74				
* SOX10 *	240	T	T240P	Waardenburg syndrome type 4C	12	0	1.25	16.25	1.00				
*SOX10*	278	I	I278V	Aganglionic megacolon	15	0	1.5	12.75	1.00	25			
*SOX10*	428	M	M428I	Abnormality of limb bone	0	0	1	15.75	0.99	14	M428I	2	22.7
*SOX10*	433	R	R433Q	not provided	6	0	1	19.5	0.99	2			
*SOX11*	417	C	C417W	not provided	4	0	1	17.25	0.95				
*SOX13*	171	S	S171L	Abnormality of the ear	36	2	1	13	0.99	2	S171L	1	26.3
*SOX13*	192	R	R192Q	Multiple	15	0	1.25	22.25	0.99	4	R192Q	5	34
*SOX13*	210	H	H210R	Neural Atrophy/Degeneration	13	0	1.5	21.25	0.98	4	H210R	4	26.7
*SOX13*	507	R	R507Q	Muscular dystrophy	16	0	1.25	19.5	0.96	3	R507Q	2	34
* SOX14 *	88	K	K88R	Abnormality of the cardiovascular system (×9)	4	0	1	23	0.95	52	K88R	8	25.8
* SOX14 *	187	T	T187K	Hypoplastic left-heart syndrome	5	0	1	21	1.00	1	T187K	2	25.4
*SOX18*	326	E	E326V	Skeletal muscle atrophy	1	0	1	10.25	1.00		E326V	1	21.4
*SOX18*	331	L	L331F	HLTS	9	0	1	11	1.00				
*SOX18*	369	L	L369V	not provided	19	0	1.25	16	1.00	1			
*SOX18*	375	A	A375T	Nephrotic syndrome	35	0	2	13.25	0.99		A375T	1	20.4

## Data Availability

The compiled dataset for SOX codons/amino acids and variants can be found at https://doi.org/10.6084/m9.figshare.21761000 (accessed on 18 December 2022). The file contains three tabs: All AAs—lists each of the SOX genes and every codon/amino acid of each along with data of conservation and variants; HMG box data—Aligned data over all SOX genes for HMG box positions; and All var AA—a list of all amino acids with variants. The molecular dynamic simulation data can be found at https://doi.org/10.6084/m9.figshare.14544339 (accessed on 18 December 2022). The SOX HMG box alignments can be found at https://doi.org/10.6084/m9.figshare.14544063 (accessed on 18 December 2022) and the phylogenetic tree at https://doi.org/10.6084/m9.figshare.14544219 (accessed on 18 December 2022).

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
