# Peer review of "Evolutionary Landscape of SOX Genes to Inform Genotype-to-Phenotype Relationships"

_genes, 2023, doi:10.3390/genes14010222_

Round 1

Reviewer 1 Report

Dear authors,

the present manuscript is an interesting study. Before accepting for publication, some aspects to be adressed:

Table 1, for some references the publication year is missing. Related to this Table will be good to explain a bit more why most of the references are before 2000, and the latest relevant are from 2011? In the last 10 years no articles related were published regarding additional disease-causing variants?

Discussions section is well organized, but missing relevant references to support most of the statements related to the current-state-of-the-art. Please improve this section accordingly.

A conclusions section is needed to highlight the novelty of thsi study and why now. Please add such section.

Author Response

We thank the reviewer for the suggestions for changes. Below is a point by point response. In addition, we performed one final editing of the manuscript through Grammarly.

 Table 1, for some references the publication year is missing. Related to this Table will be good to explain a bit more why most of the references are before 2000, and the latest relevant are from 2011? In the last 10 years no articles related were published regarding additional disease-causing variants?

Response: The dates were trimmed off a few due to a formatting issue in the table, which has been corrected. We added the following paragraph into the discussion to address why the , “Before the 2000s, most variant insights were published in literature [103], and therefore we developed a list of common SOX genetic variant papers in Table 1. This table is not meant to be all-inclusive but captures the most notable SOX genetic variant papers. Following the establishment of ClinVar and other databases, it became more typical for variants to be listed in both publications and within the database. Thus, combining the early variant manuscripts (Table 1) with the most common variant databases makes a larger map of the genomics landscape of SOX genes possible.”

Discussions section is well organized, but missing relevant references to support most of the statements related to the current-state-of-the-art. Please improve this section accordingly.

Response: We have added 10 additional citations within the discussion.

A conclusions section is needed to highlight the novelty of thsi study and why now. Please add such section.

Response: We have added the following section 5 Conclusions: “This study is the first to systematically analyze the evolutionary conservation within thousands of sequences of SOX genes/proteins with multiple database integrations of human variants linked to phenotypes. From discovering novel domains outside the HMG box, linking several SOX genes to novel phenotypes, and identifying several inherited variants linked to phenotypic traits, we show the promise of new genomic discoveries within a large transcription factor family. While we continue to advance our knowledge of SOX members, this work also highlights the importance of using paralog mapping to understand variants better, especially when occurring in a paralog family member that is lesser studied with new knowledge to be gained. Overall, this shows that even in 2023, there is much genomic knowledge to be learned, and bioinformatics holds many promises in advancing our genomic insights.”

Reviewer 2 Report

Dear Authors, 

I am very impressed with your detailed and thorough analysis of the SOX genes. I  believe that is a comprehensive and in-depth study, but also a summary that is needed in a world of growing amounts of data that require systemization and meaning. 

I have only a few comments

The first suggestion is about Figures 1 and 5. I think it would be beneficial to modify these figures to make them more readable. In Figure 1, the small font is now difficult to read, especially in the printed version. Figure 5 could be divided into two smaller ones. Parts A and D in particular are currently hard to read.

Regarding the Molecular dynamic simulation results, I also recommend showing the radius of gyration.

In Figure 7, section D, the histology annotation of variants is shown separately from the tissue/organ annotation. It seems that from a clinical point of view, it would be preferable to present one annotation for different histological subtypes of tumours of different organ origin, eg lung adenocarcinoma, lung squamous carcinoma etc. In the biology of cancer, considering a problem only by referring to the histological type of tissue from which a given tumour originates or is built, or considering only in terms of organ location, is incomplete, because different histological subtypes of tumours within only one organ may significantly differ in the nature of growth and clinical course.

Best regards

Author Response

We thank the reviewer for the suggestions for changes. Below is a point by point response. In addition, we performed one final editing of the manuscript through Grammarly.

The first suggestion is about Figures 1 and 5. I think it would be beneficial to modify these figures to make them more readable. In Figure 1, the small font is now difficult to read, especially in the printed version. Figure 5 could be divided into two smaller ones. Parts A and D in particular are currently hard to read.

Response: For figure 1 and 5 we have provided a link within the figure legend to a higher-resolution image for download. As both figures are very difficult to parse into two figures, and we do not wish to lose the information we have mapped, this seemed like the best solution to resolution issues that may arise.

Regarding the Molecular dynamic simulation results, I also recommend showing the radius of gyration.

Response: To show radius of gyration in a more useful approach for genetics folks we have made a video of SOX18 WT and the E137K movement. We added the following text to the paper, “This suggests that E137K is of functional impact. Molecular dynamic simulations of SOX18 protein show that E137K results in elevated motion of the salt bridge (E/K137 with R140, https://youtu.be/CKg3dhkRHxY), which increases the availability of charges for po-tential protein interactions to increase, suggesting it impacts protein SOX18 function.”

In Figure 7, section D, the histology annotation of variants is shown separately from the tissue/organ annotation. It seems that from a clinical point of view, it would be preferable to present one annotation for different histological subtypes of tumours of different organ origin, eg lung adenocarcinoma, lung squamous carcinoma etc. In the biology of cancer, considering a problem only by referring to the histological type of tissue from which a given tumour originates or is built, or considering only in terms of organ location, is incomplete, because different histological subtypes of tumours within only one organ may significantly differ in the nature of growth and clinical course.

Response: This was a great suggestion. We merged the two categories and reanalyzed the combined data. Additionally, we showed the top ten and listed the SOX gene with the most high impact variants for the cancer type. We added the following text, “These HMG box variants are primarily found in Adenocarcinoma and large intestine samples (Figure 7D), with many of the SOX proteins having high-risk variants for the pathology. Of the top ten cancer types, SOX9 has the leading high-risk variants for large intestine adenocarcinoma and stomach not specified (NS). SOX17 is the leading protein for skin NS, stomach adenocarcinoma, endometrium endometrioid carcinoma, lung adenocarcinoma, and urinary tract NS. SOX13 accounts for 22% of the high-risk variants in lung squamous cell carcinoma, SOX 2 for 33% of pancreas ductal carcinoma, and SOX7 for 40% of the thyroid NS.”